# Effect of CytoSorb Coupled with Hemodialysis on Interleukin-6 and Hemodynamic Parameters in Patients with Systemic Inflammatory Response Syndrome: A Retrospective Cohort Study

**DOI:** 10.3390/jcm11247500

**Published:** 2022-12-18

**Authors:** Vanja Persic, Alexander Jerman, Marija Malgaj Vrecko, Jernej Berden, Vojka Gorjup, Adela Stecher, Milica Lukic, Matjaz Jereb, Gordana Taleska Stupica, Jakob Gubensek

**Affiliations:** 1Center for Acute and Complicated Dialysis and Vascular Access, Department of Nephrology, University Medical Center Ljubljana, 1000 Ljubljana, Slovenia; 2Faculty of Medicine, University of Ljubljana, 1000 Ljubljana, Slovenia; 3Center of Intensive Internal Medicine, University Medical Center Ljubljana, 1000 Ljubljana, Slovenia; 4Department of Anesthesiology and Surgical Intensive Therapy, University Medical Center Ljubljana, 1000 Ljubljana, Slovenia; 5Department of Infectious Diseases, University Medical Center Ljubljana, 1000 Ljubljana, Slovenia

**Keywords:** hemoadsorption, cytokines, dialysis, interleukin-6

## Abstract

Excessive release of cytokines during systemic inflammatory response syndrome (SIRS) often leads to refractory hypotension and multiple organ failure with high mortality. Cytokine removal with hemoadsorption has emerged as a possible adjuvant therapy, but data on interleukin-6 (IL-6) reduction and outcomes in clinical practice are scarce. We aimed to evaluate the effect of CytoSorb hemoadsorption on laboratory and clinical outcomes in shocked patients with SIRS. We designed a retrospective analysis of all patients with SIRS treated with CytoSorb in intensive care units (ICU). IL-6, laboratory and hemodynamic parameters were analyzed at approximate time intervals during CytoSorb treatment in the whole cohort and in a subgroup with septic shock. Observed and predicted mortality rates were compared. We included 118 patients with various etiologies of SIRS (septic shock 69%, post-resuscitation shock 16%, SIRS with acute pancreatitis 6%, other 9%); in all but one patient, CytoSorb was coupled with renal replacement therapy. A statistically significant decrease in IL-6 and vasopressor index with an increase in pH and mean arterial pressure was observed from 6 h onward. The reduction of lactate became significant at 48 h. Results were similar in a subgroup of patients with septic shock. Observed ICU and in-hospital mortalities were lower than predicted by Sequential Organ Failure Assessment (SOFA) (61% vs. 79%, *p* = 0.005) and Acute Physiology and Chronic Health Evaluation (APACHE) II (64% vs. 78%, *p* = 0.031) scores. To conclude, hemoadsorption in shocked patients with SIRS was associated with a rapid decrease in IL-6 and hemodynamic improvement, with improved observed vs. predicted survival. These results need to be confirmed in a randomized study.

## 1. Introduction

Systemic inflammatory response syndrome (SIRS) is a complex pathophysiologic response to a number of infectious stimuli (pathogen-associated molecular patterns; PAMPs) due to bacterial, viral, or disseminated fungal infection and non-infectious hostile stimuli (damage-associated molecular patterns; DAMPs), produced as a consequence of trauma, ischemia, or other tissue damage (pancreatitis, burns, immune-mediated organ injury...) [1,2]. While the inflammatory and immune responses to such stimuli in the acute state of disease are usually protective and allow the host to overcome the infection and survive, in SIRS, there can be an exaggerated and dysregulated immune response that causes hyperinflammation, which often results in organ dysfunction, failure and even death [3].

Septic shock remains the most common cause of this life-threatening condition of hyperinflammation [4], where an enormous amount of relatively small (<40 kDa) cytokines, such as interleukin-6 (IL-6), IL-10, tumor necrosis factor and others are released in the blood (so-called cytokine storm) [5]. Cytokines also play a crucial role in the pathogenesis of severe acute pancreatitis [6] and in the pathogenesis of post-resuscitation shock since one of the consequences of ischemia–reperfusion syndrome after out-of-hospital cardiac arrest (OHCA) or in-hospital cardiac arrest (IHCA) is sepsis-like features with systemic inflammatory response [7,8]. These conditions all have high mortality rates [9,10], and it has been shown that high levels of cytokines correlate with worse outcomes in patients [6]. Therefore, it is not surprising that, in addition to standard treatment of these deleterious conditions, cytokine elimination with extracorporeal blood purification techniques represents an attractive adjuvant treatment option. In 90 s, a non-selective removal of a broad spectrum of inflammatory mediators (including cytokines) with continuous veno-venous hemofiltration (CVVH) in patients with septic shock and acute kidney failure was demonstrated, but without significant changes in their serum concentrations or clinical outcomes [11,12,13,14].

In the past decades, new adsorption materials were developed and tested for use in patients with severe elevation of cytokines. One of the hemoadsorption cartridges approved in Europe is CytoSorb, with a large effective surface area of approximately 40,000 m^2^. Hydrophobic molecules between 5–60 kDa in size (size range of cytokines, PAMPs, DAMPs, endogenous molecules, e.g., myoglobin or bilirubin and some drugs) can easily get captured and irreversibly bound to the inner surfaces of beads, while smaller or larger solutes, including electrolytes and larger immune cells, recirculate unchanged. The removal is size- and concentration-dependent: the higher cytokine plasma levels are, the faster and more effective reduction [15]. The removal of cytokines in SIRS represents a so-called “desirable” effect of CytoSorb treatment, but little is known of any “undesirable” effects. One of them is the removal of antibiotics [16], many of which are also removed by hemoadsorption, which could have a negative impact on patients’ outcomes since antibiotics are the most basic and effective therapy for sepsis.

Therefore, hemoadsorption has great theoretical potential in the treatment of sepsis and SIRS. To our best knowledge, only one prospective and a few large observational studies have reported the use of CytoSorb and its effect on interleukin-6 (IL-6) removal and hemodynamics (as a surrogate outcome) in patients with refractory septic shock and/or hypercytokinaemic state [17,18,19]. Until large randomized studies are performed to assess the effects on clinical outcomes, smaller studies reporting on surrogate outcomes, such as IL-6 removal and hemodynamic improvement, can also inform clinical practice. The objective of our study was to describe the time course of IL-6 as a marker cytokine during treatment with CytoSorb and the effect on clinical conditions, mainly blood pressure and vasopressor requirement, in critically ill patients with different etiology of refractory shock due to SIRS.

## 2. Materials and Methods

### 2.1. Patients and Study Design

We performed a retrospective analysis of all patients treated with CytoSorb adsorber in intensive care units of University Medical Center Ljubljana, Slovenia, from January 2017 to October 2019. The study complied with the Declaration of Helsinki (as revised in Tokyo 2013) and was approved by the National Medical Ethics Committee (No. 0120-533/2019/5); patient consent was waived due to the retrospective nature of the study.

### 2.2. CytoSorb Treatment

According to a priori agreement between nephrologists and intensivists in our institution, CytoSorb treatment was initiated (a) within 2 h in patients with suspected fulminant meningococcemia, necrotizing fasciitis, toxic shock syndrome or in post-splenectomy patients with suspected fulminant sepsis, and (b) within 6–12 h (i.e., after some “stabilization period” to allow for the standard treatment to show its effects), if the patient met one of the following criteria: refractory septic shock (see definition below) after 6 h of standard treatment (fluid resuscitation, vasoactive therapy), severe acute pancreatitis with SIRS, severe burns (third- or fourth-degree burns that cover more than 20% of total body surface [20]) with SIRS, refractory post-resuscitation shock (refractory shock after 6 h of standard therapy without myocardial dysfunction being the main contributor to shock and with elevated IL-6). We decided not to start with CytoSorb treatment in advanced medical conditions with a limited life expectancy (e.g., advanced metastatic cancer, severe neurological impairment, end-stage chronic disease) or shock duration > 48 h. Refractory shock was defined as (a) an increasing noradrenaline (NA) requirement (>0.5 μg/kg/min) to maintain mean arterial pressure (MAP) ≥ 65 mmHg, (b) serum lactate level ≥2 mmol/L despite adequate volume resuscitation and (c) multi-organ failure (at least 2 organs). SIRS was defined by meeting any two of the following criteria (a) body temperature over 38 or under 36 degrees Celsius, (b) heart rate greater than 90 beats/minute, (c) respiratory rate greater than 20 breaths/minute or partial pressure of CO_2_ less than 4.2 kPa or (d) leukocyte count greater than 12 × 10^9^/liter or less than 4 × 10^9^/liter or over 10% immature cells.

We combined CytoSorb and renal replacement therapy in all but one patient. CytoSorb cartridges were placed in a pre-filter position. CytoSorb was coupled with continuous veno-venous hemodialysis (CVVHD) in patients with severe hemodynamic instability or requirement for sustained fluid removal. CVVHD was performed with Prismaflex (Gambro, Lund, Sweden) dialysis monitors and AN 69 ST hollow fiber filters (acrylonitrile and sodium methallyl sulfonate copolymer, surface treated with polyethylene imine) using automated regional citrate anticoagulation (RCA). Blood flow was set to 150–200 mL/min, and dialysate flow was between 1500–2000 mL/h. CytoSorb was coupled with intermittent hemodialysis (IHD) in patients with severe hyperkalemia, severe metabolic acidosis or hyperammonemia. IHD was performed with AK 200 (Gambro, Lund, Sweden) dialysis monitors and polysulfone (Helixone^®^, Fresenius Medical Care, Bad Homburg, Germany) membrane filters using either RCA or unfractionated heparin (in cases of additional indication for heparin). Blood flow was set to 200–250 mL/min, and dialysate flow was between 300–500 mL/min. If electrolyte and acid-base balance were achieved, dialysate flow was stopped and only hemoperfusion was continued; nevertheless, for the purpose of removing calcium-citrate complexes and avoiding citrate accumulation, hemoperfusion was switched to hemodialysis for 2 h after every 2–3 h of hemoperfusion. CytoSorb was discontinued when the vasopressor requirement decreased below 20% of the initial dose or if the patient’s clinical condition was not improving. If deterioration of clinical status after cessation was observed, treatment with CytoSorb was recommenced.

### 2.3. Data Collection

Baseline patient demographics, laboratory values and indications for CytoSorb were collected from medical records. Indications for CytoSorb were categorized as (a) septic shock, (b) SIRS in patients with acute pancreatitis, (c) post-resuscitation shock in patients after OHCA or IHCA and (d) other. Serum levels of IL-6, lactate, pH and procalcitonin (PCT), mean arterial blood pressure (MAP) and vasoactive-inotropic score (VIS) were recorded from medical records at the beginning of CytoSorb treatment after approximately 6, 12, 24 and 48 h of the beginning of the first procedure, when available, and compared to baseline. IL-6 was measured by an electrochemiluminescence assay (Cobas e 411, Roche Diagnostics GmbH, Mannheim, Germany). Values reported as >5000 ng/L from the laboratory were analyzed as 5000 ng/L. VIS was calculated as follows: dopamine dose (μg/kg/min) + dobutamine dose (μg/kg/min) + 100 × epinephrine dose (μg/kg/min) + 100 × norepinephrine dose (μg/kg/min) + 10 × milrinone dose (μg/kg/min) + 10,000 × vasopressin dose (U/kg/min). Three-day, ten-day and twenty-eight-day mortalities were recorded, and intensive care unit (ICU) and in-hospital mortality rates were compared to predicted mortalities as calculated by modified SOFA (sequential organ failure assessment) and APACHE II (acute physiology and chronic health evaluation II) scores, which were calculated at the initiation of treatment to estimate disease severity.

### 2.4. Statistical Analysis

Basic patients’ characteristics and outcome parameters are presented as number (percent) for categorical parameters, mean ± standard deviation (SD) or median and interquartile range (IQR) for normally and non-normally distributed continuous parameters, respectively. For greater clarity, all main outcome parameters (IL-6, MAP, VIS, lactate, pH and PCT) were analyzed with nonparametric methods, although MAP was found to be normally distributed. Their change over time was analyzed by paired Wilcoxon signed-ranks test comparisons to baseline (time 0) values; due to multiple comparisons (five time points, four comparisons), a Bonferroni correction was applied, and *p* value of <0.012 (0.05/4) was considered statistically significant. Predicted ICU and in-hospital mortalities were calculated based on median SOFA [21,22] and APACHE II scores [23,24], respectively; from the predicted mortality ratio, we calculated predicted absolute number of fatalities for our cohort size. Predicted and observed mortalities were compared using Chi-squared test. Statistical significance was defined as *p* < 0.05, except in multiple comparisons, as described above. Statistical analyses were conducted in R (R Core Team, 2014) [25].

## 3. Results

### 3.1. Patients’ and Procedures’ Characteristics

We screened 147 patients and finally included 118 patients (see Figure 1) whose baseline characteristics and indications for hemoadsorption therapy are described in Table 1. In 91% of patients, CytoSorb was coupled with IHD, and only a minority (8%) received the combination of CytoSorb and CVVHD. Nineteen patients (16%) were treated with CytoSorb and IHD/CVVHD filter incorporated in parallel with the extracorporeal membrane oxygenation (ECMO) circuit. The median procedure duration was 12.0 (IQR 8.0–14.1) hours, and the median number of procedures per patient was 1 (IQR 1–2, range 1–7). RCA was used in all but one IHD procedure and in all CVVHD procedures. No device-related adverse events were observed during treatment time. Two procedures were terminated prematurely due to dialysis machine dysfunction and the risk of extracorporeal system coagulation. SOFA score, APACHE II score and VIS score at the initiation of CytoSorb therapy are presented in Table 1.

### 3.2. IL-6 and Clinical Outcomes

IL-6 values during the first 48 h after initiation of the first CytoSorb treatment are shown in Figure 2 and Table 2. Adsorption therapy was effective in reducing IL-6 levels, which already decreased significantly 6 h after initiation of treatment, with a slight increase between 12 and 24 h, likely due to the effect of CytoSorb discontinuation.

The observed reduction in IL-6 was accompanied by a favorable clinical response, indicated by a significant increase in MAP and a reduction of vasopressor therapy (VIS score) requirements from 6 h onward and a decrease in lactate levels, which became significant at 48 h. pH also increased significantly at 6 h of combined treatment with RRT and CytoSorb (Figure 3, Table 3).

### 3.3. Septic Shock Patients

In a subgroup of 68 patients with septic shock (as defined in the consensus statement [1]), we observed similar results (Figure 4, Table 4). There was a significant decrease in IL-6 levels, a significant increase in MAP and pH and a decrease of VIS 6 h after the start of CytoSorb therapy. Forty-eight hours after CytoSorb therapy, IL-6 levels remained lower compared to the values at the beginning of treatment. A continuous increase in pH was observed during treatment with RRT—CytoSorb. Lactate levels slowly decreased and were lower with borderline significance at 12 h and significantly lower after 48 h, compared to the initial values. PCT levels increased in the first 6 h; later, the decrease in PCT concentration was observed, and it became significant after 24 h of CytoSorb treatment.

### 3.4. Predicted vs. Observed Mortality

In our cohort, three-day mortality was 25%, ICU mortality was 61% and in-hospital mortality was 64%. Predicted ICU mortality, based on SOFA score, was 79% for our cohort, which was significantly higher than the observed 61% (*p* = 0.005, Table 5), and predicted in-hospital mortality, based on APACHE II score, was 78%, which was also significantly higher than the observed 64% (*p* = 0.031, Table 6). Analyses of patients’ mortality stratified by SOFA and APACHE II scores are shown in Figure 5 and Table 5 and Table 6.

## 4. Discussion

To the best of our knowledge, this is one of the largest cohort studies of CytoSorb hemoadsorption in patients with SIRS, where laboratory and clinical parameters related to shock as well as mortality were assessed. We have shown an effective reduction in IL-6 levels with corresponding clinical response, including a reduction in vasopressor therapy and an increase in MAP, suggesting the efficacy of Cytosorb treatment. The observed ICU and in-hospital mortality rates in our cohort were significantly lower than the predicted ones according to the SOFA and APACHE II scoring system, and these data support previous studies [15,26] but not all [19,27]. For example, Schittek et al., in a retrospective controlled study, did not find any reduction of ICU or hospital mortality after the implementation of hemoadsorption for patients in septic shock with acute renal failure [27]. Similarly, in a recently published analysis of the international CytoSorb registry data, a statistically significant benefit in mortality in the overall cohort was not confirmed [19]. While the majority of studies evaluated only patients with septic shock [15], patients with different causes of SIRS, such as pancreatitis or post-resuscitation shock, were included in our cohort, as cytokine storm is a common pathophysiology in these states.

In addition to measuring MAP, we performed an assessment of hemodynamic status with calculations of the vasoactive-inotropic score (VIS) during CytoSorb treatment and not only norepinephrine dosage [15] since, in the majority of patients in shock, multiple vasoactive drugs are usually used. We observed hemodynamic stabilization during and after CytoSorb treatment with a statistically significant increase in MAP and a decrease in VIS score, occurring early (within 6 h) in the course of treatment. Similar findings were described in some other smaller observational studies [15,28,29], while, on the other hand, in another study, no significant reduction in norepinephrine dosage was observed over 24 h of CytoSorb treatment [30]. Higher VIS values have been associated with worse outcomes in pediatric and adult patients [31,32,33] and are, therefore, an important surrogate outcome. Even though our set of patients had extremely high initial values of VIS (median 70, range 43–101) with a more profound hemodynamic instability in comparison to patients in Calabro et al. (VIS score of 20, range 10–35 [29]) at the beginning of adsorption treatment, mortality was similar (28-day mortality in our study 56% and 30-day-mortality 55% in the Calabro study [29]).

Since severe acidosis is associated with catecholamine-refractory hypotension and decreased myocardial contractility, the correction of acidosis per se by renal replacement therapy is beneficial for an enhanced effect of catecholamines and achievement of hemodynamic stability. One of the advantages of IHD over CRRT is the ability for a fast correction of electrolyte disbalance, acid-base status and a greater clearance of small and middle-size molecules (e.g., urea, lactate, ammonia…). In a 2019 study by Nogi et al., there was a statistically significant decrease in norepinephrine dose in patients with metabolic acidosis and septic shock treated with IHD alone, and they assumed that the improvement of circulation was probably reflected by a slight decrease in lactate levels during IHD treatment [34]. In our clinical practice, we coupled CytoSorb with standard hemodialysis high-flux filters in the majority of our patients in order to accelerate the clearance of lactate, correction of metabolic acidosis and treat the commonly present hyperkaliemia. In line with Nogi’s findings [34], we observed a statistically significant increase in pH with a reduction of lactate levels that decreased slowly and reached statistical significance 48 h after CytoSorb was started.

To make comparison with other studies easier, we analyzed a subgroup of patients with septic shock within our cohort. Our results confirm data from previous prospective studies reporting a significant reduction of IL-6 during CytoSorb procedures [17,18,19]. On the contrary, in a randomized trial, Schadler et al. were not able to detect a difference in plasma IL-6 levels between patients treated with CytoSorb for 6 h as compared to controls [18] since there was a slow and sustained reduction in plasma IL-6 in both groups. In our cohort of septic patients, IL-6 levels substantially decreased during CytoSorb treatment coupled with IHD/CVVHD. Importantly, in our group, the baseline IL-6 levels were very high compared to Schadler (5000 (800–5000) ng/L vs. 552 (162–874) ng/L), which indicates a significantly sicker population. This might have affected the CytoSorb adsorbing effectiveness, which is known to be concentration-dependent. Nevertheless, controlled studies are necessary to firmly show the significance of IL-6 adsorption on the course of IL-6 levels and clinical outcomes.

Procalcitonin (PCT) is one of the markers of bacterial infection, and its serum concentration is positively correlated with the severity of the infection [35]. PCT may also have a toxic effect on sepsis [36] since it was reported to decrease cardiovascular stability in experimental models, so active removal of PCT could be beneficial. Because of a relatively small molecular weight (approx. 13–14 kDa) PCT can be removed using high-flux dialysis filters [37] predominantly with filtration [38,39,40], but can also be eliminated by adsorption on AN69ST membranes and CytoSorb [41]. After the expected initial increase at 6 h, there was a tendency toward a decline in PCT levels, so it is possible that PCT was partially removed during the procedure. A statistically significant decrease in PCT levels observed 24 h after the CytoSorb procedure could be explained by the combined effects of removal with CytoSorb, improving the clinical condition and timely antibiotic therapy.

Our study has several limitations. The most important are the observational design, multiple simultaneous interventions (CytoSorb, dialysis…) and the lack of a control group. Therefore, it is not possible to firmly confirm any cause–effect relationship between the CytoSorb treatment, clinical improvement and mortality. Furthermore, it is known that scoring systems (SOFA and APACHE), designed in 1998 [21,22] and 1983 [23,24,42], can overestimate the mortality in contemporary cohorts. Hence, a part of the observed difference between predicted and observed mortality is likely not the effect of hemoadsorption but rather an overall improvement in the treatment of patients in the last decades. Nevertheless, the study format is in line with similar observational studies [15,17,29]. Furthermore, because of the retrospective nature of the study, there was a significant number of missing data at some time points, reducing the power of the study.

## 5. Conclusions

In conclusion, we have shown that CytoSorb treatment, used in patients with SIRS of different etiologies, is associated with a decrease in IL-6 and a corresponding beneficial effect on hemodynamics and possibly also survival. Further randomized clinical trials are needed to fully elucidate the effect of CytoSorb on the removal of cytokines and survival.

## Figures and Tables

**Figure 1 jcm-11-07500-f001:**
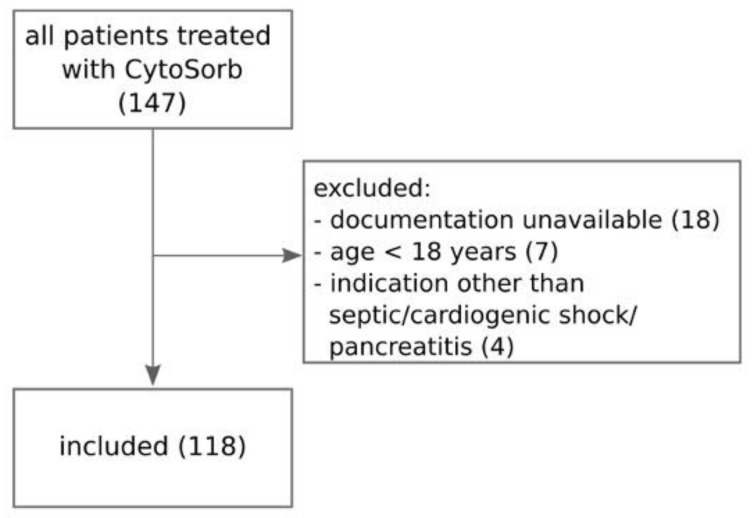
Flowchart of patient selection process.

**Figure 2 jcm-11-07500-f002:**
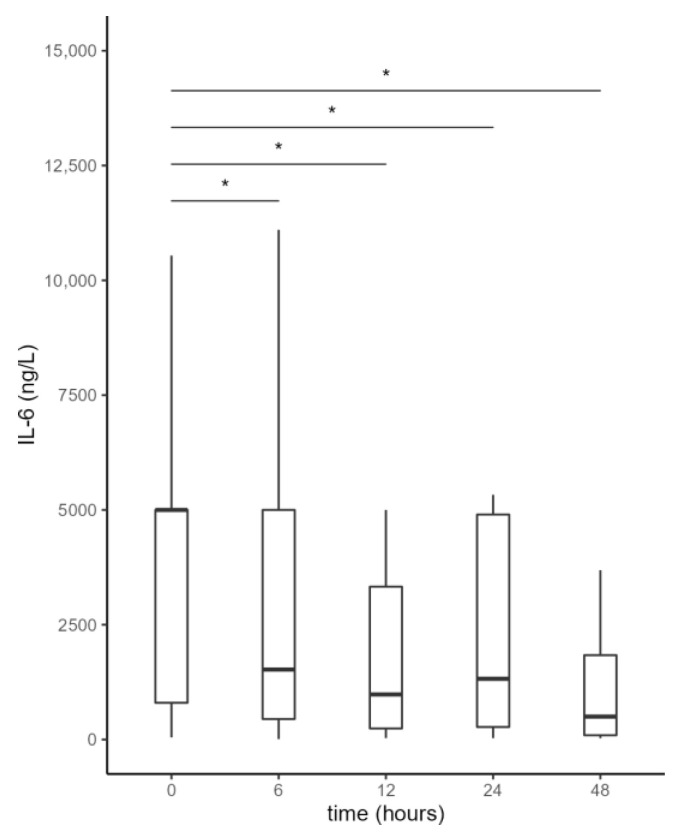
Serum interleukin-6 (IL-6) levels from the beginning of CytoSorb treatment to 48 h after the start of the first CytoSorb treatment. Values presented as median, interquartile range, minimum and maximum; not all outliers are shown. Comparison by paired Wilcoxon test. Values *p* < 0.012 were considered statistically significant and are indicated by *.

**Figure 3 jcm-11-07500-f003:**
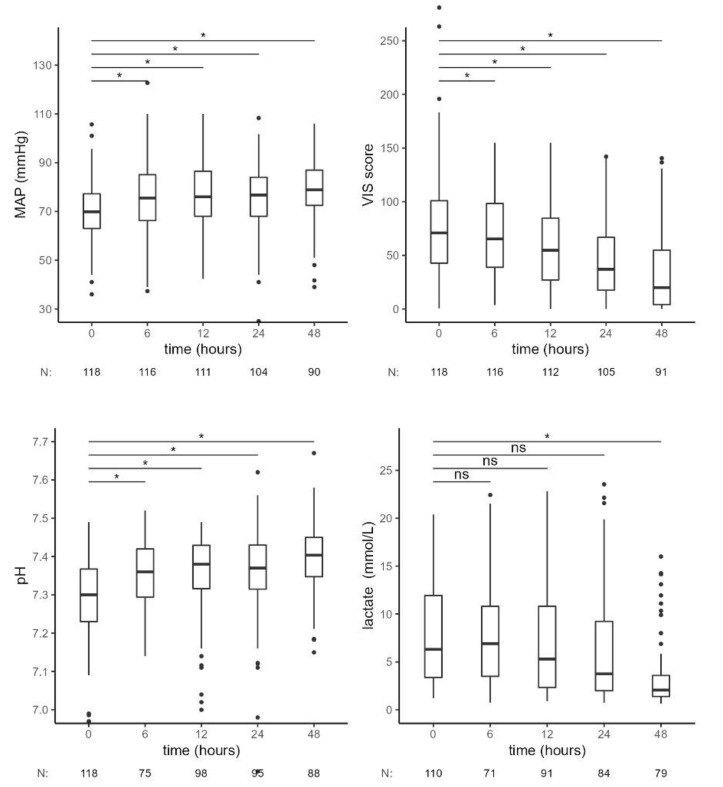
Hemodynamic and laboratory parameters from the beginning of CytoSorb treatment to 48 h after the start of the first CytoSorb treatment. Values presented as median, interquartile range, minimum and maximum; outliers and N of available measurements are shown. Comparison by paired Wilcoxon test. Values *p* < 0.012 were considered statistically significant and are indicated by *. MAP—mean arterial pressure, VIS—vasoactive-inotropic score, ns—not significant.

**Figure 4 jcm-11-07500-f004:**
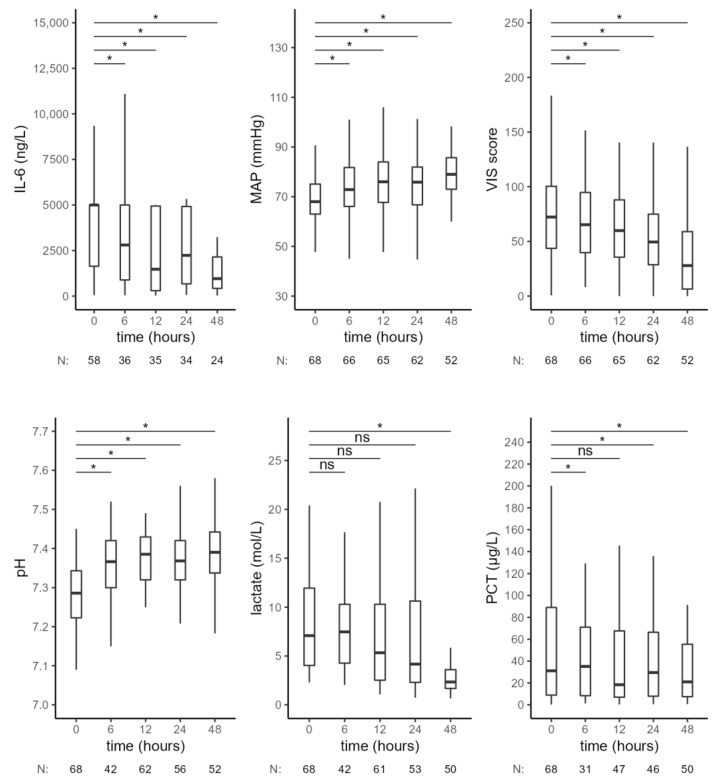
Hemodynamic and laboratory parameters from the beginning of CytoSorb treatment to 48 h after the start of the first CytoSorb treatment in a subgroup of patients with septic shock. Values presented as median and interquartile range, comparison by paired Wilcoxon test, * *p* < 0.012 was considered statistically significant. MAP—mean arterial pressure, VIS—vasoactive-inotropic score, PCT—procalcitonin, ns—not significant.

**Figure 5 jcm-11-07500-f005:**
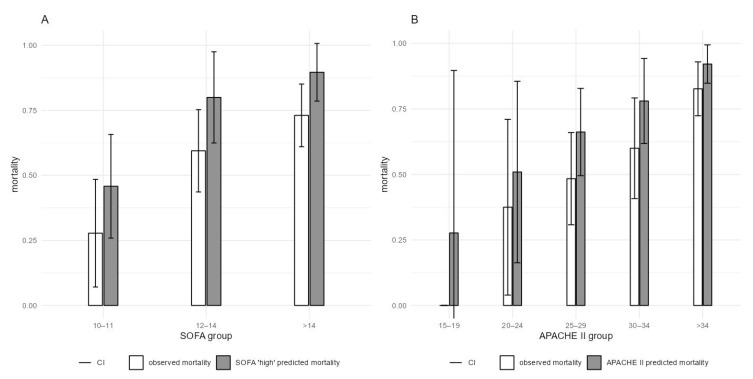
ICU and in-hospital mortality, stratified by the highest SOFA and APACHE II at the first CytoSorb treatment (**A**) Actual ICU mortality with 95% CI compared to SOFA-estimated mortality. (**B**) Actual in-hospital mortality with 95% CI compared to APACHE II-estimated mortality. CI–confidence interval.

**Table 1 jcm-11-07500-t001:** Patients’ characteristics at initiation of treatment and outcome. Data are presented as mean ± standard deviation, median [interquartile range] and number (percent) as appropriate.

Characteristics	Value
Demographics	
N	118
Age (years)	63 [51–73]
Female gender	34 (29%)
BMI	27.6 [26.0, 30.4]
Comorbidities:	
No comorbidities	30 (25%)
Hypertension	62 (53%)
Diabetes mellitus	26 (22%)
Peripheral artery disease	7 (6%)
Heart failure	8 (7%)
COPD	13 (11%)
Indication for CytoSorb treatment:	
Septic shock	81 (69%)
SIRS—after OHCA or IHCA	19 (16%)
SIRS—acute pancreatitis	7 (6%)
other	11 (9%)
Vital signs:	
MAP (mmHg)	69.6 ± 12.8
Heart rate (1/min)	105 ± 22
Organ support:	
Mechanical ventilation	114 (97%)
ECMO support	19 (16%)
Renal replacement therapy:	
intermittent hemodialysis	107 (91%)
CRRT	10 (8%)
none (pure hemoperfusion)	1 (1%)
Medications:	
Vasoactive support	118 (100%)
Hydrocortisone	90 (76%)
Antibiotic treatment	113 (96%)
Immunosuppression *	15 (13%)
Laboratory parameters:	
lactate (mmol/L)	6.3 [3.8–11.9]
IL-6 (ng/L)	5000 [800–5000]
pH	7.3 [7.2–7.4]
creatinine (µmol/L)	200 [139–308]
Disease severity scores:	
VIS	70 [43–101]
SOFA	14 [12–16]
APACHE II	33 [28–37]
Mortality	
3-day mortality	30 (25%)
10-day mortality	47 (40%)
28-day mortality	66 (56%)

BMI—body mass index, COPD—chronic obstructive pulmonary disease, SIRS—systemic inflammatory response syndrome, OHCA—out-of-hospital cardiac arrest, IHCA—in-hospital cardiac arrest, MAP—mean arterial pressure, ECMO—extracorporeal membrane oxygenation, CRRT—continuous renal replacement therapy, VIS—Vasoactive-Inotropic Score, SOFA—Sequential Organ Failure Assessment, APACHE—Acute Physiology and Chronic Health Evaluation, * chronic therapy for immune-mediated diseases.

**Table 2 jcm-11-07500-t002:** Serum interleukin-6 (IL-6) levels from the beginning of CytoSorb treatment to 48 h after the start of the first CytoSorb treatment. Values presented as median, interquartile range.

Time (h)	0	6	12	24	48
IL-6 (ng/mL)	5000 [800–5000]	1524 [446–5000]	982 [242–3329]	1323 [273–4900]	499 [94–1837]
Compared to time 0 (p)	/	<0.001	<0.001	<0.001	<0.001
Compared to previous time category (p)	/	<0.001	0.027	0.002	<0.001

**Table 3 jcm-11-07500-t003:** Hemodynamic and laboratory parameters from the beginning of CytoSorb treatment to 48 h after the start of the first CytoSorb treatment. Values presented as median, interquartile range, statistical comparison to time 0.

Time (h)	0	6	12	24	48
MAP (mmHg)	70 [63–77]	76 [66–85]	76 [68–87]	77 [68–84]	79 [72–87]
*p*-value	/	<0.001	<0.001	0.002	<0.001
VIS	71 [43–101]	65 [39–98]	55 [27–85]	37 [18–67]	20 [4–55]
*p*-value	/	0.009	<0.001	<0.001	<0.001
pH	7.30 [7.23–7.37]	7.36 [7.29–7.42]	7.38 [7.32–7.43]	7.37 [7.31–7.43]	7.40 [7.35–7.45]
*p*-value	/	<0.001	<0.001	<0.001	<0.001
s-lactate (mmol/L)	6.31 [3.38–11.94]	6.90 [3.50–10.80]	5.30 [2.33–10.81]	3.76 [2.00–9.23]	2.06 [1.40–3.58]
*p*-value	/	0.484	0.124	0.156	<0.001

MAP—mean arterial pressure, VIS—vasoactive-inotropic score.

**Table 4 jcm-11-07500-t004:** Hemodynamic and laboratory parameters from the beginning of CytoSorb treatment to 48 h after the start of the first CytoSorb treatment in a subgroup of patients with septic shock. Values presented as median and interquartile range, statistical comparison to time 0.

Time (h)	0	6	12	24	48
IL-6 (ng/mL)	5000 [1639–5000]	2804 [885–5000]	1474 [297–4943]	2236 [672–4917]	958 [422–2147]
*p*-value	/	0.002	0.003	<0.001	<0.001
MAP (mmHg)	68 [63–75]	73 [66–82]	76 [6–84]	76 [67–82]	79 [73–86]
*p*-value	/	<0.001	<0.001	0.003	<0.001
VIS	72 [44–100]	65 [40–95]	60 [36–88]	50 [29–75]	28 [6–59]
*p*-value	/	0.006	0.001	<0.001	<0.001
pH	7.29 [7.22–7.34]	7.37 [7.30–7.42]	7.38 [7.32–7.43]	7.37 [7.32–7.42]	7.39 [7.34–7.44]
*p*-value	/	<0.001	<0.001	<0.001	<0.001
lactate (mmol/L)	7.1 [4.0–12.0]	7.5 [4.3–10.3]	5.3 [2.5–10.3]	4.2 [2.3–10.6]	2.3 [1.7–3.6]
*p*-value	/	0.271	0.025	0.162	<0.001
PCT (µg/L)	31 [9–89]	35 [8–71]	18 [7–68]	30 [8–66]	21 [7–55]
*p*-value	/	<0.001	0.036	0.009	<0.001

MAP—mean arterial pressure, VIS—vasoactive-inotropic score, PCT—procalcitonin.

**Table 5 jcm-11-07500-t005:** Highest SOFA score and observed vs. predicted ICU mortality rate. Comparison for grouped ICU mortality by Chi-squared test.

SOFA Group	Mortality—Reference	Mortality—Predicted	Mortality—Observed	*p*-Value
10–11	46%	8.3/18	5/18 (28%)	/
12–14	80%	29.6/37	22/37 (59%)	/
>14	90%	46.8/52	38/52 (73%)	/
all		85/107 (79%)	65/107 (61%)	0.005

SOFA: sequential organ failure assessment, ICU - intensive care unit.

**Table 6 jcm-11-07500-t006:** APACHE II score, observed in-hospital mortality vs. predicted mortality rate. Comparison by Chi-squared test.

APACHE II Group	Mortality—Reference	Mortality—Predicted	Mortality Observed	*p*-Value
15–19	27%	1/2	0/2 (0%)	/
20–24	50%	4/8	3/8 (38%)	/
25–29	65%	20/31	15/31 (48%)	/
30–34	80%	20/25	15/25 (60%)	/
>34	89%	47/52	43/52 (83%)	/
all		92/118 (78%)	76/118 (64%)	0.031

APACHE II: acute physiology and chronic health evaluation II.

## Data Availability

The datasets used and/or analyzed during the current study are available from the corresponding author upon reasonable request.

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
