# Peer review of "Effect of CytoSorb Coupled with Hemodialysis on Interleukin-6 and Hemodynamic Parameters in Patients with Systemic Inflammatory Response Syndrome: A Retrospective Cohort Study"

_jcm, 2022, doi:10.3390/jcm11247500_

Round 1
Reviewer 1 Report
- First of all, congratulations on a well-done analysis.
- Line 54: would change "aforementioned" to "these"
- Line 61: change "important effects" to "significant changes"
- Line 61-62: recommend "or clinical outcomes" instead of "and with no clear clinical benefit"
- Line 64: change "cytokinemia" to "elevation of cytokines"
- Line 72: would add "a" to ...represents "a" so-called ...
- Line 73: would add "any" to ...is known of "any" undesirable...
- Line 82: recommend removing "Therefore"
- How much of the improvement in pH, lactate and procalcitonin is related to IL-6 hemoabsorption and how much is related to renal replacement therapy?
- any data on antibiotic removal by the hemofiltration cartridge?
- is there a difference between the patients treated with IHD and CVVH?
Reviewer 2 Report
This is an interesting manuscript that evaluate the effect of CytoSorb hemoadsorption on laboratory and clinical outcomes in patients with SIRS. The whole study is well designed despite of its nature of retrospective cohort study and the results are properly analyzed and dicussioned. I have only a few minor concerns before the acceptance of this work.
1. Please provide the information of the hemofilters used in the IHD or CVVHD sessions because the type of dialysis membrane may also have an effect on the outcomes of ciritically ill patients that receives extracorporeal hemoadsorption sessions.
2. In line 108, the authors stated that refractory shock was defined as (a) an increasing noradrenaline (NA) requirement (>0.5 μg/kg/min) despite adding corticosteroids to maintain mean arterial pressure (MAP) ≥ 65 mmHg, is there any reference that supports your statement. I donot agree your point that corticosteroids are used to maintain mean arterial pressure in septic patients.
3. In table 1, please provide BMI instead of body weight.
4. Please add the Final results of the International CytoSorb Registry (PLoS One. 2022 Oct 25;17(10):e0274315. doi: 10.1371/journal.pone.0274315. PMID: 36282800; PMCID: PMC9595535) to your discussion.
